# Liver Transplantation in Patients with Portal Vein Thrombosis: Revisiting Outcomes According to Surgical Techniques

**DOI:** 10.3390/jcm12072457

**Published:** 2023-03-23

**Authors:** Domenico Pinelli, Matteo Cescon, Matteo Ravaioli, Flavia Neri, Annalisa Amaduzzi, Matteo Serenari, Greta Carioli, Antonio Siniscalchi, Michele Colledan

**Affiliations:** 1Department of Organ Failure and Transplantation, Papa Giovanni XXIII Hospital, 24127 Bergamo, Italy; 2Hepatobiliary and Transplant Unit, Policlinico Sant’Orsola IRCCS, University of Bologna, 40138 Bologna, Italy; 3FROM Research Foundation, Papa Giovanni XXIII Hospital, 24127 Bergamo, Italy; 4Anesthesia and Intensive Care Unit, Policlinico Sant’Orsola IRCCS, University of Bologna, 40138 Bologna, Italy; 5School of Medicine and Surgery, University of Milan-Bicocca, 20126 Milan, Italy

**Keywords:** Yerdel grade, reno-portal anastomosis, porto-mesenteric bypass, spleno-portal shunt, thrombectomy

## Abstract

Surgical strategies for graft portal vein flow restoration vary from termino-terminal portal vein anastomosis to more complex bypass reconstructions. Although the surgical strategy strongly influences the post-operative outcome, the Yerdel grading is still commonly used to determine the prognosis of patients with portal vein thrombosis (PVT) undergoing liver transplantation (LT). We retrospectively reviewed the cases of LT performed on recipients with complex PVT at two high-volume transplantation centres. We stratified the patients by the type of portal vein reconstruction, termino-terminal portal vein anastomosis (TTA) versus bypass reconstruction (bypass group), and assessed a multivariable survival analysis. The rate of mortality at 90 days was 21.4% for the bypass group compared to 9.8% in the TTA group (*p* = 0.05). In the multivariable correlation analysis, only a trend for greater risk of early mortality was confirmed in the bypass groups (HR 2.5; *p* = 0.059). Yerdel grade was uninfluential in the rate of early complications. A wide range of surgical options are available for different situations of PVT which yield an outcome unrelated to the Yerdel grading. An algorithm for PVT management should be based on the technical approach and should include a surgically oriented definition of PVT extension.

## 1. Introduction

Portal vein thrombosis (PVT) is a well-recognised complication in cirrhotic patients, with a reported incidence ranging from 2% to 23% of cases [1]. PVT has always been defined as a risk factor in liver transplantation (LT) since Yerdel et al. demonstrated the direct correlation between its extension along the porto-mesenteric axis and the post-transplant mortality [2] in 2000. In the last 20 years, all the studies published on this matter have employed the Yerdel classification to stratify the severity of the clinical and surgical scenario in order to correlate it with the post-LT outcome. Although this classification is very intuitive and easy to apply, it does not reliably reflect the surgical and clinical implication of the extension of the PVT and entails some major limitations we should acknowledge. First, the determination of the PVT grading according to Yerdel is dependent on the intraoperative judgment of the surgeon, which lacks objectivity. This is particularly true when facing some grey-zone situations, as Yerdel 3 PVT cases which present with incomplete thrombosis of superior mesenteric vein (SMV) extended to jejunal branches, or partial splenic vein thrombosis. Secondly, when the thrombosis assessment is conducted pre-operatively, we should keep in mind that the CT imaging definition of PVT grading has not undergone a process of validation through a general consensus between surgeons and radiologists. Moreover, in this case, the PVT classification cannot be standardised. Finally, the choice of the surgical strategy for portal vein reconstruction is not always homogeneous in the different grades of PVT, especially in higher grades (Yerdel 3 and 4), being driven by the experience or the personal preference of the surgical team. A few other PVT classifications have been proposed with the intention to better describe the intraoperative surgical situation and potential reconstructive strategies; for example, describing the presence of spleno-portal shunts [3,4]. However, these classifications have not been prospectively validated and fully adopted in surgical studies.

Some meta-analyses have investigated the outcome of LT in the presence of PVT, taking into account more recent publications [5,6]. However, the available data are heterogeneous, and the studies generally present monocentric retrospective case series, which are often limited in number. More importantly, they often omit details about the severity of the underlying liver disease, the entity and the type of PVT extension, and the surgical technique adopted to restore the portal vein flow. Furthermore, the cause of death is often not adequately reported. All these issues impair the deduction of univocal conclusions.

A recent publication from one of the groups reported the results of our case series of LT with PVT, in which the type of portal vein reconstruction was better correlated with the post-LT outcome than the Yerdel grade [7]. In line with this hypothesis, we aimed to extend the study population of LT recipients with PVT by putting together the cases from two high-volume transplantation centres, analysing early mortality and morbidity stratified for surgical reconstructive techniques. In accordance with Banghui et al. [8], we grouped the portal vein reconstruction techniques on a “functional” basis, defining as physiological the surgical strategies that allow the drainage of splanchnic circulation and the mitigation of the presinusoidal portal hypertension. Figure 1 depicts the types of portal vein reconstruction adopted by the two centres in this study.

## 2. Materials and Methods

### 2.1. Study Design

This is a retrospective analysis of a cohort of adult patients with PVT who underwent liver transplantation at two centres in Northern Italy, Bergamo and Bologna, from January 2000 to December 2020. The study was approved by the Internal Review Board.

The purpose of this study was to assess post-transplantation morbidity, mortality, and survival of grafts and patients, stratifying by the type of portal reconstruction. We aimed to compare the early transplantation outcomes between the patients who underwent a termino-terminal portal anastomoses (TTA group), and those who underwent other physiological portal reconstructions (bypass group). After reviewing all the cases with documented portal thrombosis at the time of transplantation, we decided to exclude from the analysis the patients with minimal or partial portal vein thrombosis involving < 50% of the lumen (Yerdel 1), and the patients who underwent a non-physiological portal reconstruction (Figure 2).

### 2.2. Data Collection

Both centres reviewed the clinical data stored in the relative scientific databases. Different independent variables were analysed relating to the recipient, the donor, and the surgical setting. The recipient features were age, gender, BMI, months spent on the waiting list, type of liver disease, type of hepatitis viral infection, presence of combined other solid organ transplantation, history of past abdominal and sovramesocolic surgery, MELD score, presence of pretransplant ascites, oesophageal varices and transjugular portosystemic shunt (TIPS), and hospitalisation status. The data related to the donor were age, weight, height, BMI, gender, cause of death, type of donation (either donation after brain death (DBD) or donation after cardiac death (DCD)), presence of latent/active HBV and HCV infection, and donor risk index (DRI). The surgical factors considered were length of operation (in minutes), length of cold ischemia time (in minutes), type of graft (whole or split), intraoperative grade of thrombosis defined by surgeon according to Yerdel classification, intraoperative portal thrombectomy and use of vascular graft for portal reconstruction, type of caval, portal, arterial and biliary reconstruction, intraoperative vascular shunt ligation, and amount of infused hemocomponents. 

The considered outcomes were days of in-hospital and on-ICU stay, occurrence of primary non-function (PNF), retransplantation and its cause, portal rethrombosis and its treatment, death and its cause, incidence of early complications within 90 days from transplantation according to the Clavien classification [9]; status of the patient and the graft, and time of patient and graft survival in months.

### 2.3. Definitions

Both centres agreed on the definitions of all analysed clinical parameters. The recipient MELD score and the donor risk index were calculated using the online calculators. For the latter score, the system required us to locate the site of the organ procurement as national, regional, or local. The DRI was designed on an American model [10], and the European interpretations of the location parameter may be subjective and misleading. Since there are no indications on how to consider this variable in other non-American realities, we decided to apply a spatial criterion, defining local as the sites of procurement located within 2 h of car travel from the Transplantation Centre, regional as the donation sites in Central or Southern Italy for which the use of an aircraft was necessary, and national as those located in a country outside of Italy. 

For concerns relating to the identification of PVT, we reviewed the operatory log and, through the description of the surgeon, we assigned a grade to the thrombosis according to Yerdel definition [2].

We adopted the classification system used by Bhangui et al. [8] to define a physiological portal reconstruction as when all or part of the splanchnic venous blood could be redirected to the liver graft. The treatment groups were classified based on the type of portal vein reconstruction: termino-terminal anastomoses (TTA); anastomoses between the portal vein and the superior mesenteric vein (PV-SMV); portal reconstruction on a splanchnic varix (PV-VX); reno-portal anastomoses, which is the connection of the portal vein to the proximal left renal vein in presence of a splenorenal shunt (RPA); and cavo-portal transposition (CPT). Except for the latter group, all the others were considered physiological portal vein reconstructions.

The complications were graded according to the classification by Dindo–Clavien, excluding the cases of retransplantation, PNF, and PVT, which were all accounted for separately.

The patient survival was assessed from the date of transplantation to the date of last patient follow-up or death, whichever event came first, while the graft survival was calculated from the date of transplantation to the date of retransplantation, last patient follow-up, or patient death, whichever event came first.

### 2.4. Surgical Strategies for Portal Reconstruction

At both centres, the termino-terminal portal anastomoses is considered the gold standard for portal reconstruction. Independent of the site and extent of the thrombosis, if complete or partial thrombectomy allowed the restoration of adequate portal flow, anatomical reconstruction was accomplished either by direct anastomosis or using an interposition venous graft from the donor. In the case of an ineffective thrombectomy, or when the vessel wall appeared unsuitable for the anastomosis after portal flow restoration, a non-anatomical but physiological reconstruction was attempted. Preoperative imaging and intraoperative findings guided the choice between an SMV to PV jump graft, enlarged varix to PV anastomosis, or RPA with or without an interposition graft. RPA was performed only in the presence of a large spontaneous or surgical splenorenal shunt. In one centre, non-physiological portal reconstruction was chosen in a few selected cases, while in the other centre this option was never considered.

### 2.5. Post-Operative Anticoagulation

Continuous intravenous infusion of heparin sodium or low-molecular-weight heparin was started as soon as it was allowed by the patient’s clinical conditions and coagulation tests, with a target activated partial thromboplastin time ratio of 2. After a few days, in the absence of thrombotic complications, the heparin infusion was replaced by the subcutaneous administration of low-molecular-weight heparin once or twice a day, which was continued for at least one month. Moreover, if the platelet count was above 50,000/L, the patient received long-term oral acetylsalicylic acid (100 mg daily).

### 2.6. Portal Flow Imaging Surveillance

Doppler ultrasonography was performed once daily throughout the first post-operative week, every two days throughout the second post-operative week, and progressively less frequently thereafter. Any suspicion of PV rethrombosis was urgently investigated using contrast-enhanced computed tomography.

### 2.7. Statistical Analysis

We used descriptive statistics to summarise the characteristics of the cohort of adult patients under study. We expressed continuous variables as medians and interquartile ranges (IQRs) and categorical variables as frequencies and percentages. We stratified the characteristics of the sample according to the type of portal reconstruction (TTA versus bypass), and according to the vital status of the patients. We tested differences between strata using the Mann–Whitney test for continuous variables, and the chi-square test (or Fisher’s exact test when appropriate) for categorical variables.

Survival for the considered sample was estimated using the Kaplan–Meier method. We computed time to death as the time, expressed in days or years, between the date of liver transplantation and the date of death, and we censored the time for survivors at the last available information/contact. We analysed both 90-day mortality and overall mortality. We stratified survival curves for the type of portal reconstruction (TTA versus bypass) and, in a more granular way, according to the following groups: TTA, SMV, Varix, and RPA. We tested differences in survival between strata using the log-rank test.

We estimated the hazard ratio (HR), along with the corresponding 95% confidence interval (CI), of 90-day mortality using the Cox proportional hazards model, and, adjusting for most relevant covariates, the univariate analyses had a significant result.

For all tested hypotheses, two-sided *p*-values of 0.05 or less were considered significant. Statistical analysis was performed using STATA software, release 16.1 (StataCorp LP, College Station, TX, USA), and was carried out at the biostatistical laboratory of the Foundation for Research at Papa Giovanni XXIII Hospital in Bergamo.

## 3. Results

### 3.1. Patients’ Demographic, Clinical, and Surgical Features

From January 2000 to December 2020, 300 adult patients with non-tumoral portal vein thrombosis underwent deceased-donor liver transplantations in the two aforementioned centres. Almost half of them were excluded from our analysis due to low-grade Yerdel 1 thrombosis (*n* = 129) or the adoption of a non-physiological portal vein reconstruction (*n* = 6); the remaining 165 patients were considered for this study. The diagram flow in Figure 2 shows the steps of patient exclusion and the distribution according to the Yerdel grade of PVT.

In Table 1, we compared the main pre-operative clinical features of the recipients, the data of the donors, and the intraoperative surgical details between the two portal vein reconstruction groups (i.e., TTA vs. bypass). The two groups’ characteristics were quite homogeneously distributed, with only a lower proportion of viral and alcohol aetiology of cirrhosis in the bypass group compared to the TTA group (23.8% vs. 34.1% for viral aetiology, and 9.5% vs. 16.3% for alcohol, *p* = 0.035); a higher DRI for the bypass group (median of 1.8 vs. 1.5, *p* = 0.014); a lower proportion of Yerdel grade 2 of thrombosis in the bypass group (14.3% vs. 54.5%); a higher proportion of Yerdel grade 3 and 4 of thrombosis in the bypass group (50.0% vs. 40.7% for grade 3 and 35.7% vs. 4.9% for grade 4, *p* = <0.001); and a higher percentage of vascular graft use in the bypass group (69.0% vs. 2.4%, *p* < 0.001). Notably, pre-operative TIPS had been placed in 8.9% and 4.8% of patients, respectively, in the TTA and bypass group. The decision to perform a bypass in patients with TIPS was driven by different factors: thrombosis of the TIPS without possibility of intraoperative recanalisation; injury to the portal vein in the attempt of removing the metallic wires; or thickening of the portal vein walls due to local inflammation.

### 3.2. Post-Trasplant Outcomes

We then used an univariate analysis to compare the post-transplant outcomes between the TTA and bypass groups. Most of the analysed outcomes were similar between the two groups except for the 90-day mortality, which was 21.4% in the bypass group and 9.8% in the TTA group (*p*-value = 0.05) (Appendix A and Table 2).

### 3.3. Early Mortality

The 90-day survival of the entire cohort of patients was of 87%, ranging from 79% in the bypass group to 90% in the TTA group (Appendix A). 

A univariate analysis was performed to assess which peri-operative factors were associated with 90-day mortality (Table 2). Among the patients who experienced an early death, compared to the survivors, there was a higher proportion of the use of vascular graft (38.1% vs. 16.7%, *p* = 0.02); non-direct duct-to-duct biliary reconstruction (19.0% vs. 16.7% for biliary-intestinal derivation and 14.3% vs. 0.7% for external diversion, *p* = 0.005); and a higher amount of red blood cell transfusion (median of 2240 mL vs. 1000 mL, *p* = 0.003). In addition, we found a significant centre effect in the 90-day mortality, with most of the dead recipients afferent to centre 1, compared to surviving patients.

Table 3 shows the multivariable Cox model results. Overall, patients who underwent bypass reconstruction, compared to TTA, had an increased risk of mortality, with an HR of 2.52 (0.96–6.58; *p* = 0.059). Stratifying by centre, the HR for bypass, compared to TTA, was 3.64 (1.14–11.64; *p* = 0.029) for centre 1, and 1.12 (0.20–6.43; *p* = 0.897) in centre 2.

A more granular KM analysis of the different groups of bypass techniques showed better short-term survival for portal reconstruction with pericholedochal or gastric varices (over 90% at 90 days) compared to renoportal and superior mesenteric vein anastomosis, which entailed a 90-day survival of 70% and 64%, respectively (Appendix A).

The details of the 21 patients deceased within 90 days from transplantation are depicted in Table 4. Twelve of the recipients belonged to the TTA group and nine to the bypass group. The median recipient age was 53 (48–59), and the median MELD was 15 (13.7–22). The Yerdel grade was 2 and 3 in nine cases (43%), and grade 4 in three cases (9%). The transplantation was a whole graft in all but one case, and in three cases, they needed a retransplantation: one for hepatic artery thrombosis and two for PNF. The median time from LT to death was 21 days (IQR 3–37). The causes of death were various: five septic shock, five cerebral events, four cardiovascular events, two haemorrhagic shock, two multiorgan failure, one primary graft non-function, and one acute antibody-mediated rejection.

### 3.4. Long-Term Survival

Figure 3 displays the Kaplan–Maier survival curves. The overall survival at 5 years for the entire population under study was 74%, ranging from 71% for the bypass group to 75% for the TTA group. This difference was not statistically different (*p* = 0.949). The majority of events in the bypass group occurred within the first 6 months, while they were more equally distributed over the study period in the TTA group.

## 4. Discussion

Intraoperative management of PVT continues to represent a major challenge in liver transplantation, as its presence still impairs the outcome of the procedure, despite the surgical and anaesthesiologic advances that have occurred in the last two decades [11]. In 2000, Yerdel showed the direct association between extension of PVT along the longitudinal–splanchnic axis and the post-operative risk of mortality [2]. Since the publication of that benchmark paper, transplantation surgeons have always relied on the simplicity of this grading system to stratify the patients with PVT at greater risk of ominous outcome after LT. Other PVT classifications have been released aiming fora more precise description of the thrombotic extension in the splanchnic district, the assessment of porto-systemic shunts, and generally with the intention to predict the feasibility of a straightforward porto-portal anastomosis over a more complex surgical reconstruction of the portal flow [3,4]. Authors are now questioning whether the Yerdel grading, and the other classifications, still intercept the different shades of complexity of these patients and reliably predict their outcomes [12]. Although many recent meta-analyses acknowledge the prominent role of the surgical strategy adopted for portal flow restoration in driving the post-transplantation outcome, the Yerdel classification still represents the starting point for the newly proposed algorithms [8]. In 2019, Bhangui et al. reviewed this topic and proposed a guide to multidisciplinary decision making before and during LT in patients with diffuse PVT. While the differentiation between physiological and non-physiological portal vein reconstruction is very useful and appropriate, the algorithm that links portal vein reconstruction to the Yerdel grade once again does not show its usefulness in clinical surgical practice because it is not necessarily connected to the post-LT outcome. The definition of complex and non-complex PVT seems surgically uneven. In our study, the complex PVT group includes challenging but infrequent scenarios (9% of cases; 27/300), most of them solved with physiological reconstructions (78%, 21/27). The non-complex PVT group includes most frequent scenarios (91% of cases; 273/300), always solved with physiological reconstructions. Most importantly, in this group, the Yerdel grade did not always dictate a priori the physiological reconstruction that would be performed. In 20% of our population, a different surgical strategy was adopted compared to what was proposed in the algorithm by Bhangui et al. Among these cases, 10% of Yerdel 2 PVT were treated with a bypass strategy as direct thrombectomy and termino-terminal anastomosis was not feasible. In 30% of Yerdel 3 PVT, a bypass technique was adopted but did not involve the use of SMV, the first choice suggested in Bhangui’s algorithm. In 30% of Yerdel 4 PVT cases, a termino-terminal porto-portal anastomosis was performed after an extended thrombectomy with recanalisation of the superior mesenteric vein. Our experience perfectly shows how a definition of surgical complexity based on Yerdel grade is not accurate and does not reflect the real surgical challenges found intraoperatively that can drive the choice among different reconstructive techniques. The conception of our study comes from the will to define how the surgical strategy on the portal vein reconstruction relates to the post-operative outcome, irrespective of the Yerdel grade. We considered all physiological strategies of portal vein reconstruction proposed by Bhangui and grouped them as termino-terminal anastomosis (TTA group) or any other bypass technique, which included the connection of the graft PV to the SMV, to pericholedocal varices, or to the renal vein in the presence of spontaneous or surgical splenorenal shunts (bypass group) [8]. Non-physiological techniques of portal flow restoration have represented a strategy for portal flow restoration in LT since Tzkais et al. first described cavo-portal transposition in a clinical series in 1998 [13], and their use is still reported in some cases [14]. However, non-physiological techniques of portal vein reconstruction yield poor outcomes, both in the short and long term. The use of such strategies should be considered on a case-by-case basis, endorsing multivisceral transplantation whenever possible, in highly specialised centres. Therefore, our choice was to avoid the inclusion of such cases in our analysis. To the best of our knowledge, the present study is the first to tackle the issue of PVT from a purely surgical perspective, which is objective, not liable to interpretation biases, and involves a robust population of patients with PVT, for whom the portal flow restoration was achieved through a variety of different techniques. We decided to focus the analysis on the more severe grades of PVT, for which more complex techniques of portal flow restoration may have been needed, in order to keep the number of patients with TTA similar and comparable to those who had received other types of portal reconstructions. In this subset of patients, we confirmed the good long-term outcome witnessed by other reports in the last two decades [15,16]; the overall 5-year survival was over 70%, which largely justifies the access to LT for this category of patients. The most important observation from our study was the identification of the bypass technique as a risk factor for mortality within 90 days, which the Yerdel grade did not provide. This finding confirmed the result from previous works [7,16,17].

In the whole case series presented here, we report an excellent 3-month survival rate, reaching 90% in the TTA group. The patients with more complex portal reconstructions had a lower but acceptable 3-month survival rate of 70%. However, we must acknowledge the role of intercentre variability. While in the TTA group, we had similar 90-day survival in the two centres, in the bypass group, this short-term survival fluctuated between 60% and 80% in the two centres. Among the bypass techniques of portal reconstruction, we found that the one burdened by a greater degree of complications was the use of small mesenteric veins, while the anastomosis to pericholedochal varices was the most successful one. This result was surprising, as the thin walls of the periportal varices make the anastomosis technically more demanding and more prone to bleeding or thrombosis. Of course, the bypass on the SMV presents different challenges, as its central position makes its isolation and dissection for anastomosis at risk of bleeding and damage to the surrounding vessels and organs. When looking in depth at the causes of early deaths, it is difficult to relate most of these events to the surgical procedure. Notably, most of the early deaths in the bypass group were due to cerebral events (four out of nine), while the causalities in the TTA group had a more heterogeneous nature. We may speculate that the clinical/surgical condition of the patients may have strongly contributed to the early deaths we observed. In these patients, MELD is not a reliable surrogate of the operative risk, and therefore, we lack a patient-oriented, pre-operative predictive factor of mortality. From this perspective, the effect of a more challenging surgical strategy, with the consequent increased length of operation and bleeding, may be more relevant for a frail patient. In the post-operative period, the strategy of administration of anticoagulant therapy to preserve the portal vein flow may have played a role.

The present study confirms the weakness of the Yerdel grading system in discriminating the complexity of portal vein thrombosis. The surgical strategy adopted can vary widely from termino-terminal porto-portal anastomosis to complex bypass vascular reconstruction, with or without vascular graft interposition. In this bicentric study, we confirmed that the adoption of a more complex portal reconstructive strategy can represent a risk factor for short-term mortality. In the last few decades, the type of surgical strategy adopted is less and less related to the extension of PVT. Therefore, it is of paramount importance to redirect the efforts of the transplantation community towards a more surgery-oriented definition of the PVT extension and severity. The evolution in imaging diagnostics has made the intraoperative unexpected finding of a PVT an unlikely event. A commonly agreed definition of PVT severity based on pre-operative imaging would allow not only the stratification of patients at risk for post-LT mortality, but also more precise planning of the surgical strategy for portal flow restoration, particularly in cases of LT from a living donor.

## Figures and Tables

**Figure 1 jcm-12-02457-f001:**
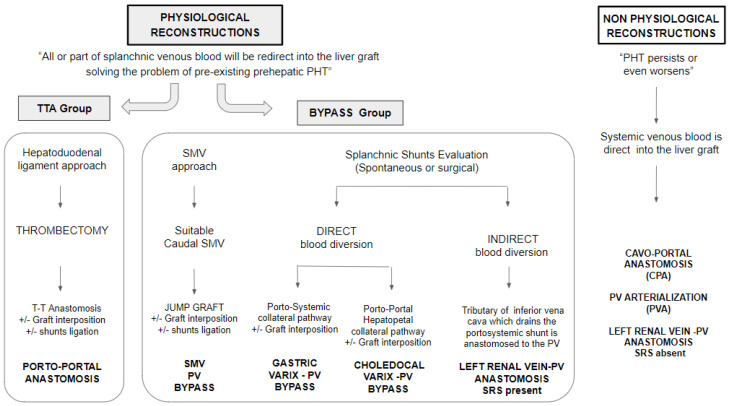
Classification of different portal reconstruction techniques according to the type of splanchnic venous blood diversion.

**Figure 2 jcm-12-02457-f002:**
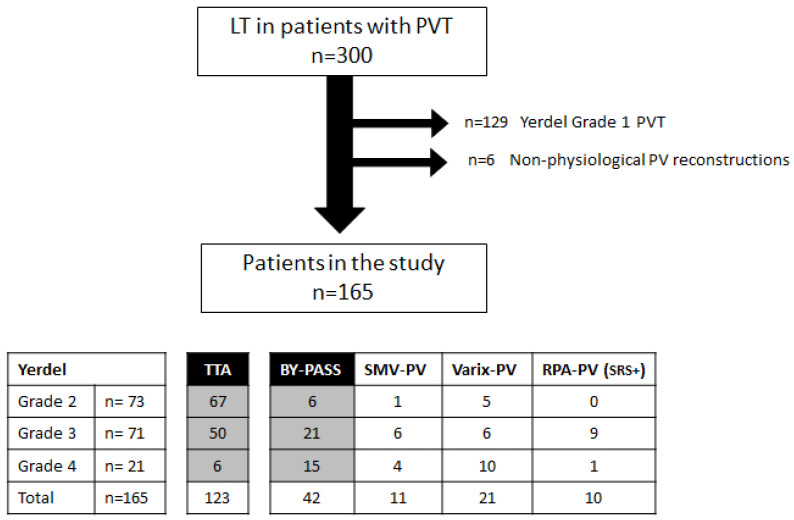
Diagram flow with the population of the study.

**Figure 3 jcm-12-02457-f003:**
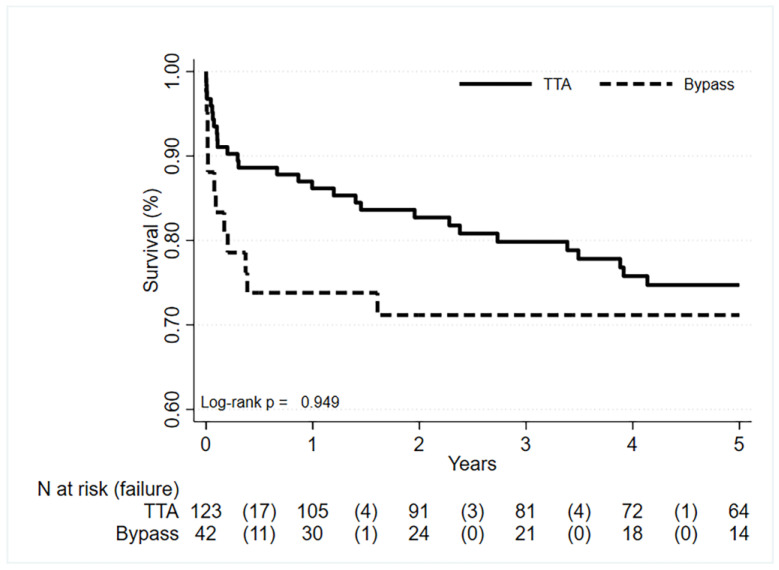
Kaplan–Meier survival analysis of the two groups TTA and bypass.

**Table 1 jcm-12-02457-t001:** Preoperative clinical features of the recipient, donor and intraoperative data compared between the two groups of portal reconstruction (TTA versus Bypass). Univariate analysis.

	Overall (165)	TTA (123)	Bypass (42)	*p*-Value
Centre; n (%)				0.90
- 1	76 (46.1)	57 (46.3)	19 (45.2)	
- 2	89 (53.9)	66 (53.7)	23 (54.8)	
**RECIPIENT**				
Male gender; n (%)	119 (72.1)	91 (74.0)	28 (66.7)	0.36
Age (ys) at transplantation; median (IQR)	56 (49–61)	56 (49–61)	54.6 (47.8–61)	0.37
BMI; median (IQR)	25.1 (23–28.3)	25.4 (23–28.7)	24.7 (227–27.5)	0.17
Waiting time (mo) in list; median (IQR)	7 (2.6–19.6)	6.8 (2.6–17.1)	9.9 (2.8–25.6)	0.15
Liver disease; n (%)				0.035
- Viral	52 (31.5)	42 (34.1)	10 (23.8)	
- Alcohol/Mixed	24 (14.5)	20 (16.3)	4 (9.5)	
- HCC	62 (37.6)	47 (38.2)	15 (35.7)	
- Other	27 (16.4)	14 (11.4)	13 (31)	
Pre-LT positioning of TIPPS; n (%)	13 (7.9)	11 (8.9)	2 (4.8)	0.52
Pre-LT MELD; median (IQR)	18 (14–25)	18 (14–25)	17 (14–23	0.58
Pre-LT ascites; n (%)	95 (57.6)	68 (55.3)	27 (64.3)	0.31
Pre-LT oesophageal varices; n (%)	147(90.2)	107 (88.4)	40 (95.2)	0.25
Hospitalization at moment of LT; n (%)	57 (34.8)	45 (36.6)	12 (29.3)	0.39
**DONOR**				
Age (ys); median (IQR)	63.0 (50.0–74.7)	61.0 (49.0–73.0)	71.2 (51.6–75.0)	0.088
BMI; median (IQR)	25.3 (23.4–27.7)	25.0 (23.4–27.7)	25.4 (23.5–27.8)	0.60
Male gender; n (%)	89 (53.9)	71 (57.7)	18 (42.9)	0.095
Cause of death; n (%)				0.25
- CVA	113 (68.5)	82 (66.7)	31 (73.8	
- Post-anoxic encephalopathy	15 (9.1)	10 (8.1)	5 (11.9)	
- Trauma	27 (16.4)	24 (19.5)	3 (7.1)	
- Other	10 (6.1)	7 (5.7)	3 (7.1)	
HBcAb positivity; n (%)	30 (18.2)	25 (20.3)	5 (11.9)	0.22
HCV positivity; n (%)	6 (3.6)	5 (4.1)	1 (2.4)	1
Donor risk index; median (IQR)	1.7 (1.4–1.9)	1.5 (1.4–1.9)	1.8 (1.5–1.9)	0.014
**INTRAOPERATORY VARIABLES**				
LT length in min; median (IQR)	425.0 (370–500)	420 (360–490)	447.5 (390–540)	0.091
CIT in min; median (IQR)	400 (335–465)	395 (331–465)	400 (350–480)	0.55
Yerdel grade; n (%)				<0.001
- 2	73 (44.2)	67 (54.5)	6 (14.3)	
- 3	71 (43)	50 (40.7)	21 (50)	
- 4	21 (12.7)	6 (4.9)	15 (35.7)	
Right split graft; n (%)	5 (3)	4 (3.3)	1 (2.4)	1
Caval anastomosis; n (%)				0.36
- Piggy back	97 (58.8)	69 (56.1)	28 (66.7)	
- Conventional	65 (39.4)	52 (42.3)	13 (31.0)	
- Latero-lateral	3 (1.8)	2 (1.6)	1 (2.4)	
Use of vascular graft for portal anastomosis; n (%)	32 (19.4)	3 (2.4)	29 (69.0)	<0.001
Biliary anastomosis; n (%)				0.083
- Duct to duct	133 (80.6)	102 (82.9)	31 (73.8)	
- Hepatico-jejunal	28 (17.0)	20 (16.3)	8 (19.0)	
- Other	4 (2.4)	1 (0.8)	3 (7.1)	

TTA: termino-terminal anastomosis; ys: years; mo: months; LT: liver transplantation; CVA: cardiovascular accident; min: minutes; CIT: cold ischemia time; TIPSS: Transjugular intrahepatic portosystemic shunt.

**Table 2 jcm-12-02457-t002:** Preoperative parameters associated to early mortality (within 90 days from liver transplantation) at the univariate analysis.

	Overall (165)	Alive at 90 Days (144)	Deceased at 90 Days (21)	*p*-Value
Portal vein reconstruction; n (%)				0.050
- TTA	123 (74.5)	111 (77.1)	12 (57.1)	
- Bypass	42 (25.5)	33 (22.9)	9 (42.9)	
Specific of portal vein reconstruction; n (%)				0.026
- TTA	123 (74.5)	111 (77.1)	12 (57.1)	
- SMV-PV jump graft	11 (6.7)	7 (4.9)	4 (19)	
- Splanchnic varices-PV	21 (12.7)	19 (13.2)	2 (9.5)	
- RPA with SRS-PV	10 (6.1)	7 (4.9)	3 (14.3)	
Use of vascular graft for portal anastomosis; n (%)	32 (19.4)	24 (16.7)	8 (38.1)	0.02
Biliary anastomosis; n (%)				0.005
- Duct to duct	133 (80.6)	119 (82.6)	14 (66.7)	
- Hepatico-jejunal	28 (17.0)	24 (16.7)	4 (19.0)	
- Other	4 (2.4)	1 (0.7)	3 (14.3)	
Amount of intraoperative red blood cells transfusions in ml; median (IQR)	1000 (520–1750)	1000 (500–1680)	2240 (1300–3055)	0.003
Centre; n (%)				0.043
- 1	76 (46.1)	62 (43.1)	14 (66.7)	
- 2	89 (53.9)	82 (56.9)	7 (33.3)	

TTA: termino-terminal anastomosis; SMV: superior mesenteric vein; RPA: reno-portal anastomosis in presence of SRS; IQR: interquartile range.

**Table 3 jcm-12-02457-t003:** Multivariate analysis of correlation between 90-day mortality and the parameters found significant at the univariate analysis and clinically relevant.

	Overall	Stratified Centre 1 (14 Events)	Stratified Centre 2 (7 Events)
Portal reconstruction			
- TTA	1 (ref)	1 (ref)	1 (ref)
- Bypass	2.52 (0.96–6.58)	3.64 (1.14–11.64)	1.12 (0.20–6.43)
	*p*-value = 0.059	*p*-value = 0.029	*p*-value = 0.897
Yerdel grade			
- II	1 (ref)	1 (ref)	1 (ref)
- III and IV	0.91 (0.34–2.44)	0.82 (0.26–2.62)	1.14 (0.20–6.51)
	*p*-value = 0.849	*p*-value = 0.735	*p*-value = 0.887
Centre			
- 1	1 (ref)		
- 2	0.39 (0.15–1.01)	NA	NA
	*p*-value = 0.051		

The data are shown as HR (95% CI) and *p*-value.

**Table 4 jcm-12-02457-t004:** Description of the main clinical and operative features of the recipients who died within 90 days with the main post-operative details.

	Recipient Age	Meld	Yerdel Grade	Graft Type	PV Anastomosis	RE-LT	Days LT-Death	CAUSE OF DEATH
**BYPASS**	46	14	3	Split (I + IV–VIII)	Gastric varix	0	0	Intraoperative cardiac arrest due to massive hemorrage
	44	44	3	whole	SMV (with VG)	0	2	PNF
	56	13	2	whole	Gastric varix	0	5	Septic shock (intestinal ischemia)
	59	15	2	whole	SMV (with VG)	0	5	Septic shock
	50	8	3	whole	RPA (with VG)	0	28	Intracranial hemorrage
	53	33	3	whole	RPA (with VG)	0	62	Cerebral Cryptoccosis
	66	14	3	whole	RPA	0	74	MOF
	59	18	4	whole	SMV (with VG)	0	6	Intracranial hemorrage
	51	17	4	whole	SMV	1 (PNF)	33	Intracranial hemorrage
**TTA**	59	15	2	whole	TTA	0	0	Intraoperative cardiac arrest
	48	15	2	whole	TTA	1 (PNF)	2	Intraoperative cardiac arrest (at the end of surgery)
	56	14	2	whole	TTA	0	3	Heart failure and pulmonary hypertension
	60	15	2	whole	TTA	0	21	Intracranial hemorrage
	53	11	3	whole	TTA	0	23	Acute AMR
	47	na	2	whole	TTA	0	27	MOF
	60	25	3	whole	TTA	0	37	Septic shock
	47	21	2	whole	TTA	0	1	PNF
	56	12	3	whole	TTA	1 (HAT)	73	Hemorragic Pancreatitis
	65	25	2	whole	TTA	0	16	Septic shock
	44	9	4	whole	TTA	0	38	Septic shock
	48	42	3	whole	TTA	0	39	Hemorragic shock (splenic aneurism rupture)

PV: portal vein; RE-LT: retransplantation; VG: vascular graft; PNF: primary graft non function; MOF: multi-organ failure; AMR: antibody-mediated rejection; HAT: hepatic artery thrombosis.

## Data Availability

The data that support the findings of this study are available from the corresponding author upon reasonable request.

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
