# Peer review of "Liver Transplantation in Patients with Portal Vein Thrombosis: Revisiting Outcomes According to Surgical Techniques"

_jcm, 2023, doi:10.3390/jcm12072457_

Round 1

Reviewer 1 Report

The present manuscript reports a retrospective bicentral study that focuses on evaluating the outcome of liver transplantation in patients with portal vein thrombosis depending on the extension of the portal vein thrombosis (Yerdel) and the surgical approach to portal vein reconstruction (T-T anastomosis vs bypass group) . The authors conclude that the Yerdel classification is a poorer survival predictor than the type of surgery.  This is an interesting study that can supports previous evidence in the field that physiological termino terminal portal anastomoses improve outcome in liver transplantation. There are some points that should be clarified to better understand the results: 

- It seems that the Yerdel classification was only based on the surgical report. Have the authors compared the classification by the surgeon and the extension by CT scan? 

-"Univariate analysis performed to assess peri-operative factors associated to mortality show higher proportion of use of vascular graft". Does this correlate with the detection of post-liver transplantation thrombosis detected by daily ultrasound? There is no data regarding the results of the daily US.

- In this line, the authors report that patients were followed this daily doppler US during the first week and that were under anticoagulation during 1 month. Was this performed according to clinical guidelines? Or is there strong evidence to support this practice? Was this done the same way in all patients during the 20 years that lasts the study?

- Did the authors find any difference in the speed of resolution of ascites or portal hypertension complications in patients where renoportal anastomosis or other bypass techniques were used? 

Author Response

  • It would be very interesting to assess whether the imaging characteristics of thrombosis matches the intraoperative findings. However, we did not have access to all the CT scans, especially the oldest cases, and could not perform a reliable comparison.
  • The higher mortality in the group where vascular grafts were used probably reflects the surgical technique of bypass more than an intrinsic risk factor related to the use of the vascular grafts. In fact, the use of vascular graft was specific of the bypass group and the rate of rethrombosis in the bypass group versus the TTA group was not significantly different (Supplementary table 1).
    The US screening is performed routinely for all the liver transplants. The cases depicted in the study underwent the same procedures, without specific assessment of vascular flow; therefore we did not have specific hemodynamic data at US to show except the incidence of thrombosis.
  • As far as we know the management of anticoagulation in recipients of liver transplant has never been clearly defined by guidelines and strong recommendations lack in this regard. We developed an internal protocol of anticoagulation as we considered the pre-transplant portal vein thrombosis a risk factor for post-operatory rethrombosis. The management of such a protocol has not changed through the years as it has always been based on heparin infusion initially and then switched to low-molecular-weight subcutaneous heparin as soon as the patient was stable and haemorrhagic complications had been excluded.
  • The long-term outcome of the different techniques of portal reconstruction was not the specific aim of our study and therefore details about the recovery from signs of hypertension were not collected. Although a direct comparison between the different techniques is lacking we do not expect to find huge differences in terms of development and resolution of ascites. There are several factors which can influence this outcome, as the pre-operative presence of ascites, the intra and post-operatory anaesthesiologic management of amines and fluid therapy and not secondarily the morpho-histological characteristics and the size match of the liver graft.

Reviewer 2 Report

This is an excellent report describing outcomes of liver transplantation in recipients with portal vein thrombosis in a large series of liver transplants at two high volume centers. The authors describe the various methods of establishing portal venous inflow into the graft and report a higher mortality in patients who required a bypass using a conduit. They discuss the shortcomings of the Yerdel classification of portal vein thrombosis. This manuscript merits publication.

I would recommend adding some operative photographs and diagrams to describe the techniques used.

I noticed that some of the patients in the bypass group had TIPS in place preoperatively. What was the indication to perform bypass in these cases? Was the portal vein damaged or not suitable for suturing?

Some of the excess mortality was due to neurological events. Were these hemorrhagic events (which may have been increased by the anticoagulation protocol) or ischemic events?

Author Response

  • Yes, often the pre-operative positioning of TIPS causes local inflammation and thickening of the portal vein walls. This is generally not a contraindication to its direct suturing to the portal. In the our cases the reason for choosing the bypass technique in the event of previous TIPS positioning were several. Sometimes the TIPS was thrombosed and could not be recanalized, in other cases the portal wall had been damaged by the metallic wires and a direct suture was not safe or feasible.
  • The deaths due to neurological events were mainly haemorrhagic in 4 cases and infective in 1 (Table 4); although an in-depth analysis of the specific anticoagulation risk in such patients we suspect that the pro-haemorrhagic diathesis need to be considered.

Round 2

Reviewer 1 Report

The authors have addressed all my concerns